# Understanding Endothelial Dysfunction and Its Role in Ischemic Stroke After the Outbreak of Recanalization Therapies

**DOI:** 10.3390/ijms252111631

**Published:** 2024-10-29

**Authors:** Patricia de la Riva, Juan Marta-Enguita, Jon Rodríguez-Antigüedad, Alberto Bergareche, Adolfo López de Munain

**Affiliations:** 1Department of Neurology, Donostia University Hospital, Dr Begiristain sn., 20003 San Sebastian, Spain; patricia.delariva@gmail.com (P.d.l.R.); abergareche@gmail.com (A.B.); adolfojose.lopezdemunainarregui@osakidetza.eus (A.L.d.M.); 2Ictus, Biogipuzkoa Institute, Doctor Begiristain sn., 20003 San Sebastian, Spain; 3Facultad de Ciencias de la Salud, Deusto University, Mundaiz 50, 20003 San Sebastian, Spain; 4Red de Investigación Cooperativa Orientada a Resultados en Salud (RICORS)-Ictus, Instituto Salud Carlos III, 28029 Madrid, Spain; 5Movement Disorders Unit and Institut d’Investigacions Biomediques-Sant Pau, Hospital Sant Pau, 08025 Barcelona, Spain; jrodriguezmu@santpau.cat

**Keywords:** stroke, endothelium, recanalization, nitrous oxide

## Abstract

Despite recent advances in treatment options, stroke remains a highly prevalent and devastating condition with significant socioeconomic impact. Recanalization therapies, including intravenous thrombolysis and endovascular treatments, have revolutionized stroke management and prognosis, providing a promising framework for exploring new therapeutic strategies. Endothelial dysfunction plays a critical role in the pathophysiology, progression, and prognosis of stroke. This review aims to synthesize the current evidence regarding the involvement of the nitric oxide (NO)/endothelium pathway in ischemic stroke, with a particular focus on aging, response to recanalization therapies, and therapeutic approaches. While significant progress has been made in recent years in understanding the relationship between endothelial dysfunction and stroke, many uncertainties persist, and although treatments targeting this pathway are promising, they have yet to demonstrate clear clinical benefits.

## 1. Introduction

Ischemic stroke (IS) is one of the leading causes of global disability and mortality [1]. Despite advances in recent decades that have improved stroke care and reduced morbidity and mortality rates, these rates remain alarmingly high. Combined with the high prevalence and incidence of this disease, IS remains a major concern for healthcare systems worldwide [1].

According to the most recent Global Burden of Disease (GBD) estimates, in 2021, stroke was the third most common GBD level 3 cause of death (7.3 million [95% UI 6.6–7.8] deaths; 10.7% [9.8–11.3] of all deaths), and the fourth most common cause of DALYs (160.5 million [147.8–171.6] DALYs; 5.6% [5.0–6.1] of all DALYs). In 2021, there were 93.8 million (89.0–99.3) prevalent and 11.9 million (10.7–13.2) incident strokes [1].

IS is defined as a neurological deficit resulting from the sudden interruption of blood flow in cerebral vessels. Its clinical presentation and progression depend on various pathophysiological factors that converge at the time of the stroke, some of which are related to blood components and their interaction with vessel walls [2].

Blood vessel walls are composed of three layers: the adventitia, the tunica media, and the endothelium, which provides a frictionless pathway for blood circulation [3].

The endothelium plays a pivotal role in maintaining vascular homeostasis in both the cerebral and systemic arterial circulation. Dysfunction or damage to the endothelial layer is correlated with conditions such as atherosclerosis, hypertension, and other cardiovascular diseases [4].

Recanalization therapies, including the intravenous administration of fibrinolytic agents and endovascular thrombectomy, are the most effective proven treatments for IS. Endothelial dysfunction can negatively affect the success of recanalization therapies, which are used to restore blood flow in stroke, by increasing reperfusion injury, compromising the blood–brain barrier, impairing vasodilation, promoting a prothrombotic state, and enhancing inflammatory responses [5].

The purpose of this review is to summarize the existing evidence regarding the involvement of the endothelium in ischemic stroke (IS), highlighting its potential positive and negative effects during the different phases of the disease. Additionally, it explores the relationship between endothelial involvement and the use of recanalization therapies and reviews the various therapeutic approaches that have been tested in this context.

## 2. The Endothelium and Its Components

The endothelium is a thin layer of cells lining the inner surface of blood vessels, including arteries, veins, and capillaries [6]. It serves not only as an anatomical barrier between blood and tissues but also functions as an endocrine organ with various roles [6].

The main components of the endothelium are:

### 2.1. Endothelial Cells

Endothelial cells (ECs) are ubiquitous within the circulatory system and are histologically distinguished by their characteristic cobblestone shape and single-cell layer arrangement. ECs have average dimensions of 30–50 μm in length, 10–30 μm in width, and 0.1–1 μm in height. They are surrounded by smooth muscle cells, or pericytes, which provide structural support to the vessels [3].

Endothelial cells can produce vasodilator factors like nitric oxide (NO) through endothelial nitric oxide synthase (eNOS) and pro-inflammatory factors such as granulocyte-macrophage colony-stimulating factor (GM-CSF), granulocyte colony-stimulating factor (G-CSF), and VEGF. They also produce anti-inflammatory molecules and extracellular vesicles [7]. While ECs exhibit remarkable tolerance to hypoxia, they are highly susceptible to reperfusion injury upon the restoration of blood flow [3].

New ECs are generated through the simple duplication of existing endothelial cells. These cells not only repair and renew the lining of established blood vessels but also create new blood vessels in a process called angiogenesis [8]. Similarly, endothelial progenitor cells (EPCs) are circulating cells capable of differentiating into mature endothelial cells. [9]. Various tissue growth factors, such as fibroblast growth factor 2 (FGF2), vascular endothelial growth factor (VEGF), and bone morphogenetic protein 4 (BMP4), mediate angiogenesis and the re-reendothelialization of injured vessels.

Functionally, ECs play a pivotal role in regulating the exchange of molecules and cells into and out of the bloodstream. They are involved in maintaining vascular structure, the coagulation system, the inflammatory response, and thrombosis. Additionally, the endothelium modulates vascular tone by regulating vascular smooth muscle cell function and producing either vasodilation or vasoconstriction through the release of nitric oxide (NO) [4].

### 2.2. Nitric Oxide

Nitric oxide (NO) is often considered a toxic gas in nature. However, 40 years ago it was discovered to function as a signaling molecule, particularly in the endothelium, where it acts as a primary neurotransmitter. NO is synthesized from L-arginine through the mediation of the NOS enzyme, with tetrahydrobiopterin (BH4) required as a cofactor. Once produced, NO activates guanylyl cyclase, which leads to the production of cGMP and the subsequent activation of various proteins essential for brain function. The fast metabolism and brief half-life of NO make it challenging to measure in clinical environments. Nitrite and nitrate, the byproducts of NO autoxidation, have been utilized as indicators of NO production [10].

Nitric oxide (NO) regulates vascular smooth muscle tone and inhibits platelet aggregation, leukocyte adhesion, and vascular smooth muscle cell growth [11]. It is widely acknowledged that the loss of endothelial NO is a central mechanism in the pathogenesis of endothelial dysfunction (ED). In both cerebral and peripheral vasculatures, reduced availability of endothelial NO leads to significant detrimental alterations in vascular functions [12].

Reduced endothelial NO availability has been demonstrated to encourage a proinflammatory and prothrombotic endothelial phenotype. NO bioavailability is closely linked to cardiovascular health, and it has been proposed that variations in NO bioavailability may help differentiate individuals with good vascular health—who are less likely to experience cardiovascular events—from those with poor vascular health [13].

### 2.3. ENOS Enzyme

NO is synthesized from L-arginine by nitric oxide synthase, and there are three different types of NO synthases: neuronal (NOS1 or nNOS), endothelial (NOS3 or eNOS), and inducible (NOS2 or iNOS). Unlike NOS1 and NOS3, NOS2 is not constitutively expressed in most tissues but is induced in response to various inflammatory and immune stimuli [14]. Inducible NOS (iNOS)-derived NO and neuronal NOS (nNOS)-derived NO play neurotoxic roles, whereas endothelial NOS (eNOS)-derived NO plays a neuroprotective role in acute IS. The toxic effects of NO produced by iNOS and nNOS are primarily due to the production of nitrates and the release of free radicals. [14]. Under physiological conditions, eNOS is responsible for the production of most endothelium-derived NO. For this reason, it plays a pivotal role in cardiovascular homeostasis, as demonstrated by clinical and experimental studies [15,16]. Phosphorylation of eNOS at serine 1177 in humans is a major regulator of eNOS activity [17].

This enzyme is coded by the NOS3 gene, which is located in the 7q35–7q36 region of chromosome 7. The NOS3 gene spans 21–22 kb and contains 25 introns and 26 exons, encoding an mRNA of 4052 nucleotides [18]. The NOS3 gene exhibits a variety of polymorphic sites, including single nucleotide polymorphisms (SNPs), variable number of tandem repeats (VNTRs), microsatellites, and insertions/deletions. To date, over 1700 genetic variations in the human NOS3 gene have been documented in the SNP database. These variations are known to influence NOS3 regulation, subsequently affecting nitric oxide (NO) production [19]. Not only individual polymorphisms but also haplotypes, which are combinations of multiple polymorphisms, have demonstrated functional effects on NOS3 expression [20].

The SNP rs2070744, also known as g.-786T>C, is one of the most extensively studied polymorphisms. Luciferase reporter gene assays have shown that this polymorphism reduces NOS3 transcriptional activity [21,22]. This effect is likely due to the increased binding affinity of a gene repressor protein called replication protein A1 (RPA1) to the NOS3 promoter when the C allele is present. RPA1 is a single-stranded DNA-binding protein involved in various aspects of DNA metabolism, including replication, repair, and recombination [23] (Figure 1). Indeed, inhibition of RPA1 expression using antisense oligonucleotides restored transcriptional activity in the NOS3 promoter containing the C allele, while RPA1 overexpression produced the opposite effect [23]. Consistent with these findings, in vivo studies have shown a tendency for lower levels of circulating NO-related markers in individuals carrying the C allele compared to those with the T allele, further supporting the functional role of this SNP [24].

Other polymorphisms reported to be functional are rs1799983, VNTR in intron 4, and SNP rs3918226 [25,26,27].

**Figure 1 ijms-25-11631-f001:**
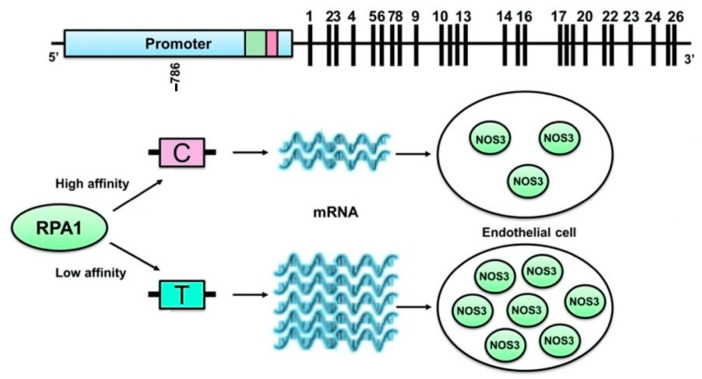
Functionality of the g.-786T>C polymorphism is mediated by RPA1, a gene repressor protein that binds to NOS3 promoter with more affinity when the C allele is present. Adapted from Oliveira-Paula [28].

The eNOS is activated by various physiological mediators, including mechanical shear stress, estrogens, insulin, and acetylcholine, among others. Post-translationally, eNOS undergoes phosphorylation at several serine (S) and threonine (T) residues, including T497, S617, S635, and S1177. Of these, phosphorylation at the S1177 residue is the most crucial mechanism for the physiological regulation of NO production [17].

### 2.4. Other Participants in the Endothelial Function

The endothelium also produces a wide range of molecules with opposing properties, such as procoagulant and anticoagulant, inflammatory and anti-inflammatory, fibrinolytic and antifibrinolytic, as well as oxidizing and antioxidizing agents.

These molecules include endothelin, von Willebrand factor, soluble cellular adhesion molecules, thrombomodulin, selectins, plasminogen activator inhibitor, angiotensin II, prostacyclins, platelet endothelial cell adhesion molecule 1, platelet-activating factor, homocysteine, tissue factor, tissue factor pathway inhibitor, monocyte chemoattractant protein, endothelial-derived hyperpolarizing factor, thromboplastin, paraoxonase, C-type natriuretic peptide, thromboxane A2, reactive oxygen species, endothelium-derived contracting factor, and cyclooxygenases, among others [4].

L-arginine serves as the substrate for NO synthesis mediated by nitric oxide synthase (NOS). In this context, arginine derivatives are considered markers of endothelial function and include homoarginine (hArg), asymmetric dimethylarginine (ADMA), and symmetric dimethylarginine (SDMA). Both ADMA and SDMA reduce the bioavailability of NO. In contrast, hArg not only acts as a substrate for NOS but also increases the availability of arginine to the NOS enzyme by inhibiting the enzyme arginase [29].

### 2.5. Cerebral Endothelium

The endothelial function varies across the arterial system, with endothelial cells being specialized to meet the specific demands of the tissues they serve. Among these, the cerebral endothelium is particularly specialized, as it plays a critical role in the formation and maintenance of the blood–brain barrier. (BBB) [30].

The BBB is a highly specialized and protective interface that separates the circulatory system from the brain’s extracellular fluid. It is essential for maintaining the unique microenvironment necessary for proper brain function. The BBB is primarily composed of endothelial cells (ECs), which are supported by pericytes, astrocytes, neural processes, and the extracellular matrix, collectively forming the neurovascular unit (NVU). The endothelial cells of the BBB are distinct in that they lack fenestrations, possess tight junctions, exhibit minimal pinocytotic activity, and express various enzymes capable of degrading both harmful and therapeutic molecules [30].

Figure 2 is a schematic illustration depicting the endothelium and the NO/NOS pathway under physiological and post-ischemic conditions.

## 3. Endothelial Dysfunction (ED)

The term ED refers to a pathological condition characterized by the impaired anticoagulant and anti-inflammatory functions of the endothelium, as well as disruptions in the modulation of vascular growth and regulation of vascular structure. A common initial mediator of ED is the impaired expression and activity of endothelial nitric oxide synthase (eNOS), leading to reduced availability of nitric oxide (NO). In the cerebral circulation, endothelial dysfunction is a prevalent feature in cerebrovascular diseases, significantly contributing to their pathophysiology, clinical impact, and prognosis [12].

Endothelial function can be evaluated using various methods that examine different vascular beds. The fundamental principle of these assessments is to determine how arteries respond to pharmacological or mechanical stimuli. Vasoactive agents such as acetylcholine, salbutamol, and bradykinin are used to measure nitric oxide (NO)-dependent vasodilation, while adenosine, nitroglycerin, dipyridamole, nitroprusside, and papaverine are employed to assess endothelium-independent vasodilation [31,32].

In peripheral circulation, flow-mediated dilation (FMD) of the brachial artery is a validated and widely used technique for assessing nitric oxide (NO)-dependent endothelial function [33]. This non-invasive method evaluates the ability of arteries to release endothelial NO in response to reactive hyperemia, which occurs following a 5-min occlusion of the brachial artery using a blood pressure cuff [32]. A comprehensive description of the FMD technique can be found in the consensus guidelines established by the International Brachial Artery Reactivity Task Force [34]. FMD is typically reduced in patients with atherosclerosis and other vascular risk factors, has been linked to an increased risk of developing new vascular events, and shows improvement with therapies targeting these risk factors [35,36].

Studies of microvascular function often involve the use of disposable, modified plethysmographic probes placed on the index fingers to measure the digital pulse waveform before and after occlusion of the brachial artery [37]. The values obtained pre- and post-occlusion are used to calculate the reactive hyperemia index (RHI), which has been shown to have a linear relationship with the occurrence of cardiovascular events [38].

In the context of cerebral circulation, endothelial function can be assessed by measuring cerebrovascular reactivity (CVR), which is defined as the percentage increase in cerebral blood flow (CBF) in response to a vasodilatory stimulus. Several techniques are available to evaluate changes in CBF, including positron emission tomography (PET), single-photon emission computed tomography (SPECT), xenon-computed tomography (Xe-CT), dynamic perfusion computed tomography, magnetic resonance imaging (MRI) with dynamic susceptibility contrast, arterial spin-labeling (ASL), and transcranial Doppler (TCD) [39]. Among these methods, SPECT, PET, and Xe-CT provide quantitative and regionally specific CBF data, but they are expensive and not widely accessible. In contrast, TCD is a non-invasive, low-cost bedside test that is widely available.

There is currently no consensus on the gold standard for assessing endothelial function in the cerebral circulation. L-arginine, a precursor of NO, can induce endothelium-dependent vasodilation and increase CBF, as measured by TCD [40]. Acetazolamide, a carbonic anhydrase inhibitor that slowly penetrates the blood–brain barrier, acts as a cerebral vasodilator. The acetazolamide test measures peak CBF augmentation 10–15 min after intravenous bolus administration, with a CBF increase of less than 10% or an absolute change of less than 10 mL/100 g/min considered pathological [39]. NG-monomethyl-L-arginine (L-NMMA), a non-selective inhibitor of all nitric oxide synthase (NOS) isoforms, is expected to decrease CBF following its administration [41].

A study comparing systemic endothelial function in hypertensive patients using FMD, intima-media thickness (IMT), and cerebral endothelial function found a correlation between IMT and the response to L-arginine (L-Arg), but not between L-Arg and FMD. This suggests that cerebral endothelial function is highly specific and, to some extent, independent of systemic endothelial function [42].

In coronary arteries, the assessment of endothelium-dependent vasodilation is typically conducted angiographically, using Doppler flow measurements to evaluate the response to endothelium-dependent agonists, primarily acetylcholine [32].

Another approach to indirectly evaluate endothelial function involves measuring molecules that are believed to play a key role in this process. These molecules include various blood biomarkers and specific cell subtypes, such as ischemia-modified albumin, pentraxin-3, E-selectin, angiopoietin, endothelial cell-specific molecule 1, arginine derivatives, von Willebrand factor, endothelial microparticles, and endothelial progenitor cells, among others [4].

## 4. Endothelial Function and Stroke

### 4.1. Physiopathology of Stroke and Endothelial Function

Stroke occurs as a consequence of disrupted blood flow in cerebral blood vessels. The endothelium, a critical component of these vessels, plays a pivotal role in the physiopathology of stroke.

Nitric oxide release following acute cerebral ischemia can have both beneficial and detrimental effects. Increased eNOS activity post-stroke appears to be neuroprotective. NO, released by eNOS, exerts a potent vasodilatory effect, inhibits platelet aggregation, and reduces leukocyte–endothelial adhesion. In support of this, animal models of stroke lacking eNOS expression (eNOS knockout models) exhibit significantly larger infarcts [43]. Conversely, administering NO donors, such as L-arginine or sodium nitroprusside, in these models results in smaller cerebral infarcts [44]. However, high concentrations of NO produced by neuronal nitric oxide synthase (nNOS) or inducible nitric oxide synthase (iNOS) have neurotoxic effects through both direct and indirect mechanisms. nNOS expression increases rapidly following an ischemic event, whereas iNOS expression rises several hours later [45]. Detrimental NO produced by nNOS and iNOS quickly reacts with superoxide to form peroxynitrite, a potent oxidant that is highly cytotoxic. Furthermore, NO promotes the release of iron from cellular stores, enhances iron-mediated lipid oxidation, and contributes to cellular energy depletion by disrupting mitochondrial enzymes and nucleic acids. The release of NO itself can also trigger neuronal apoptosis [46]. In line with these findings, nNOS-deficient mice subjected to cerebral ischemia-reperfusion show smaller infarcts and experience higher relative cerebral blood flow after reperfusion, suggesting a role for nNOS in early BBB disruption [47].

Additionally, the inflammatory response following a stroke triggers the release of cytokines, including tumor necrosis factor-α (TNF-α), interleukins (ILs), and interferon-γ (IFN-γ). This cytokine release leads to endothelial activation, which is characterized by increased vascular permeability, impaired NO signaling, and subsequent vasomotor dysfunction [48]. Under conditions of heightened oxidative stress, a deficiency in essential co-factors, particularly tetrahydrobiopterin (BH4), shifts the production of superoxide instead of NO—a process known as eNOS uncoupling. This uncoupling contributes to ED [48].

### 4.2. Endothelial Dysfunction in Ischemic Stroke

#### 4.2.1. The Role of Endothelial Dysfunction in Stroke Initiation

Endothelial dysfunction in stroke leads to oxidative stress, inflammation, increased vascular tone, BBB damage, and further cerebral complications. ED is also a key feature in chronic conditions such as atherosclerosis and hypertension. It is well established that endothelial function is impaired early in the development of atherosclerosis, even before morphological changes occur—a process closely linked to stroke [49]. Studies have shown that patients with traditional vascular risk factors for stroke, such as arterial hypertension, diabetes, and smoking, exhibit impaired systemic endothelial function [50,51].

A cohort study of older adults followed for five years found that FMD levels correlated with the occurrence of vascular events globally, and specifically with stroke events [36].

In the context of genetics and stroke, research examining the association between eNOS polymorphisms and ischemic stroke (IS) risk has yielded conflicting results, which appear to vary across different ethnic groups [52,53].

Regarding the assessment of ED in stroke, systemic ED—characterized by reduced forearm vasodilation in response to acetylcholine [54] and cerebral ED, evidenced by decreased cerebrovascular reactivity to L-arginine, are both observed in patients who have had a stroke [55]. Peripheral ED, assessed through the reactive hyperemia index, has also been identified in patients who have had a stroke and is correlated with ADMA levels [56]. Furthermore, serum levels of endothelial markers and adhesion molecules, including plasma activity of von Willebrand factor, serum levels of thrombomodulin, and plasma concentrations of P-selectin and E-selectin, are elevated in the acute phase of stroke, indicating endothelial cell activation [57].

Specific studies comparing cerebral versus systemic ED in patients who have had a stroke are limited. However, in patients with arterial hypertension, no correlation has been found between cerebral and systemic ED [42].

#### 4.2.2. Endothelial Dysfunction in Stroke Subtypes

Regarding stroke subtypes, several studies have suggested an association between lacunar IS and ED. Both systemic and cerebral endothelial function are more impaired in patients with lacunar stroke compared to healthy controls and patients with similar vascular risk factors [58,59]. ED, assessed by FMD and reactivity to L-arginine, has also been reported in patients who have had a stroke with carotid stenosis, with improvements observed months after endarterectomy [49,60]. Interestingly, patients with atrial fibrillation showed significantly better FMD results than those without it, suggesting that ED may not be directly related to cardioembolic stroke [61].

Additionally, individuals with leukoaraiosis exhibit significant impairments in both cerebral (cerebrovascular reactivity to L-arginine) and systemic (FMD) endothelial function, with the severity of ED correlating with the extent of leukoaraiosis [62]. Patients with leukoaraiosis also present the highest levels of serum markers of endothelial activation compared to other stroke subtypes [63].

Cerebral autosomal dominant arteriopathy with subcortical infarcts and leukoencephalopathy (CADASIL) is an autosomal dominantly inherited cause of stroke [64]. In CADASIL, degeneration of vascular smooth muscle cells occurs while the endothelium remains morphologically normal [65]. Patients with CADASIL demonstrate reduced baseline CBF and impaired hemodynamic reserve due to endothelium-independent vascular dysfunction. This was suggested by a study that found decreased dermal blood flow response to capsaicin, a molecule that induces relaxation of vascular smooth muscle cells in the skin, while no ED was detected through FMD, a method that assesses NO-dependent vasodilation [66].

Overall, the role of the endothelium may vary across different stroke etiologic subtypes. However, an important question arises as to whether these associations merely reflect the influence of vascular risk factors in patients who have had a stroke rather than being independently related to the disease and its specific subtypes as has been proposed.

#### 4.2.3. Endothelial Dysfunction and Stroke Severity and Prognosis

Regarding stroke severity and prognosis, a study involving 120 patients who have had a stroke found that FMD levels negatively correlated with stroke severity, with median FMD levels being lower in patients with poor outcomes [67]. A subsequent study by the same authors reported a correlation between systemic endothelial dysfunction, as assessed by FMD, and the risk of vascular events during follow-up [68].

Similarly, increased NO metabolites in cerebrospinal fluid (CSF) were associated with greater brain injury and early neurological deterioration in patients who had a stroke [69].

Table 1 provides a summary of the most commonly used techniques to assess systemic and cerebral endothelial function, along with studies conducted in patients who have had a stroke.

Regarding arginine derivatives, ADMA and SDMA are associated with an increased risk and incidence of ischemic stroke (IS), and their elevation post-stroke contributes to secondary brain injury. In contrast, H-Arg is inversely associated with adverse events and mortality in cerebrovascular diseases and may represent a modifiable protective factor [29].

Endothelial progenitor cells (EPCs) also appear to play a significant role in stroke. Vascular risk factors are linked to a decrease in EPC levels [70], and patients who have had a stroke exhibit lower numbers of EPCs at stroke onset compared to healthy controls [71]. Furthermore, it has been observed that in response to acute ischemic stroke (AIS), the body increases circulating EPC levels by mobilizing these cells from their niches [72,73]. Higher levels of circulating EPCs are associated with better prognosis in AIS, with one study showing that patients with higher numbers of circulating EPCs had smaller ischemic lesions and a higher likelihood of neurological improvement and favorable outcomes at 7 and 90 days [74]. Additionally, in patients undergoing carotid endarterectomy, EPC levels have been shown to play a crucial role in carotid plaque growth and vulnerability [75].

The unique characteristics of the cerebral endothelium have been proposed as a potential underlying reason for the increased incidence of cerebrovascular complications associated with COVID-19 infection [76]. Additionally, a subset of patients with post-COVID-19 syndrome exhibit diminished reactive hyperemia index (RHI) and altered endothelial biomarkers, suggesting that endothelial dysfunction may play a significant role in the pathophysiology of this condition [77].

Overall, ED is a critical factor in the physiopathology of stroke. It is associated with an increased risk of stroke, with this correlation varying among different stroke subtypes, and provides valuable information about stroke outcomes and prognosis.

Given this, could the assessment of endothelial function be utilized in personalized medicine to enhance risk profiling in patients with ischemic stroke, thereby complementing traditional vascular risk factors?

### 4.3. Reperfusion Therapies and Endothelial Dysfunction

Currently, reperfusion therapies, including the intravenous administration of fibrinolytic agents and endovascular thrombectomy, are the most effective treatments for stroke. However, approximately half of the patients receiving these therapies do not achieve a definitive clinical benefit and are not functionally independent 90 days post-stroke [78]. Hemorrhagic transformation (HT) is more likely with recanalization treatments and is known to contribute to a worse stroke prognosis [79].

Endothelial dysfunction could play a role in the effectiveness and outcomes of recanalization therapies. First, when blood flow is restored to previously ischemic tissue, it can trigger reperfusion injury, which is characterized by inflammation and oxidative stress. Endothelial dysfunction can exacerbate this injury by contributing to the breakdown of the blood–brain barrier, increased production of ROS, and heightened inflammatory responses. This can worsen tissue damage even after successful recanalization [80]. Second, a healthy endothelium maintains the integrity of the BBB, and when ED is present, the BBB may be compromised, increasing the risk of HT and cerebral edema following recanalization [81]. Third, in ED, reduced NO bioavailability can impair vasodilation, limiting the effectiveness of recanalization therapies as adequate NO levels are necessary for optimal blood flow restoration to ischemic regions. Fourth, ED is associated with a prothrombotic state, increasing the risk of reocclusion after recanalization [5]. Fifth, thrombus removal using stent retrievers is associated with gadolinium vessel enhancement and BBB disruption in about 50% of patients, suggesting that local endothelial damage increases with the number of stent passes and the use of alteplase [82]. And last, half of the thrombi retrieved from patients treated with stent retrievers contain endothelial cells, indicating endothelial damage secondary to direct mechanical injury from thrombectomy [83].

The role of the endothelium’s condition prior to recanalization therapies rather than after in modulating the response to these treatments remains unclear.

It is also worth noting that arginine is listed as one of the ingredients in the commercial formulations of alteplase and tenecteplase, Actylise* and Metalyse*, the two widely used intravenous thrombolytic agents in ischemic stroke. Although the exact amount of arginine in each product is not disclosed by the manufacturer, this component could theoretically enhance nitric oxide (NO) bioavailability and amplify the beneficial effects of these drugs through a mechanism distinct from their thrombolytic properties. However, this remains speculative, as no specific research has been conducted to confirm this hypothesis.

In summary, ED can negatively affect the success of recanalization therapies by increasing reperfusion injury, compromising the blood–brain barrier, impairing vasodilation, promoting a prothrombotic state, and enhancing inflammatory responses.

### 4.4. Collateral Response to Stroke and Endothelial Dysfunction

Stroke induces focal hypoperfusion, prompting the brain to activate compensatory mechanisms aimed at preserving blood flow in ischemic tissue. One such mechanism is the recruitment of collateral blood vessels [84].

Collateral circulation has been shown to influence the prognosis of patients who have had a stroke. The presence of robust collateral vessels is associated with higher recanalization rates, smaller infarct sizes, and better clinical outcomes in IS patients, both with and without reperfusion therapies [85,86]. However, the collateral response varies among individuals, and the specific factors contributing to this variability are not fully understood, though it is believed that the endothelium may play a significant role.

The 786T>C NOS3 polymorphism, previously mentioned, has been linked to poorer collateral circulation in coronary arteries [87]. However, no studies have yet assessed the relationship between this or other NOS3 polymorphisms and cerebral collaterals in stroke.

Interestingly, rapamycin has been shown to increase collateral perfusion and reperfusion cerebral blood flow in both Wistar rats and comorbid spontaneously hypertensive rats. This effect appears to be mediated by enhanced eNOS activation, as it was blocked by the non-specific NOS inhibitor L-NAME [88].

### 4.5. Blood–Brain Barrier Disruption, Endothelium, and the Risk of Hemorrhagic Transformation After Stroke

Hemorrhagic transformation (HT) is a common and natural consequence of infarction, likely driven by multiple factors [79]. Structurally, evidence suggests that the primary mechanism leading to blood extravasation is the disruption of the BBB. During cerebral ischemia, astrocytes express NOS, which contributes to the formation of peroxynitrite, leading to BBB breakdown, vasogenic edema, and subsequent HT [81]. However, no clinical studies have specifically examined the role of the NO/NOS pathway in HT following stroke, whether treated with recanalization therapies or not.

Interestingly, a study from our laboratory found that carriers of the SNP rs2070744 T allele had a higher risk of hemorrhagic transformation after endovascular therapy in patients with acute IS [89]. One possible explanation for this finding is that T allele carriers exhibit lower affinity for RPA1, leading to reduced inhibition of the eNOS gene promoter and consequently higher levels of NO. Since NO is a potent vasodilator, this increased vasodilation could potentially facilitate hemorrhagic transformation after recanalization. However, this hypothesis remains to be tested. In this context, higher NO levels could be associated with treatment complications and worse clinical outcomes.

### 4.6. Age, Stroke, and Endothelial Cell Senescence

The increased prevalence of cardiovascular disease with advancing age is partly due to endothelial aging, also known as “endothelial senescence”. Endothelial cell (EC) senescence is a pathophysiological process characterized by structural and functional changes in the endothelium, leading to increased intercellular permeability, arterial stiffness, impaired angiogenesis and vascular repair, and dysregulation of vasodilation and vasoconstriction responses.

This age-related process is mediated by various molecules, including sirtuins and klotho. Sirtuins are a group of silent information regulator 2 proteins with nicotinamide adenine dinucleotide (NAD^+^)-dependent deacetylase and ADP-ribosyltransferase activities, playing roles in DNA repair, EC senescence, cell cycle regulation, and overall organism longevity. Klotho is an anti-aging gene, and its protein expression in plasma is reduced in conditions such as coronary artery disease, stroke, heart failure, and peripheral arterial disease. Genetic and epigenetic changes contributing to EC senescence include telomere shortening, DNA methylation, and histone acetylation [90] (Figure 3).

Elderly individuals, compared to younger subjects, exhibit impaired CBF, which seems to be dependent on the NO pathway. A study found that the infusion of L-NMMA, an NOS inhibitor, significantly decreased CBF and increased CVR in older subjects but had no effect on CBF in younger participants. This suggests that the regulation of CBF in the elderly may rely more heavily on the integrity of the NO pathway [91].

EC senescence plays a critical role in aging-induced vascular dysfunction, contributing to the initiation, progression, and advancement of cardiovascular diseases. Specifically, stroke incidence increases dramatically with age, and age is a major prognostic factor in stroke [90]. EC senescence may therefore underlie the increased incidence and poor prognosis of stroke in older patients. However, no studies to date have specifically investigated this hypothesis.

## 5. Therapeutic Approaches Regarding Endothelial Dysfunction and Stroke

Given the involvement of the endothelium and its components in stroke pathophysiology and prognosis, the NO-endothelium pathway has been considered a promising therapeutic target in stroke management.

These therapeutic strategies encompass a combination of preventive measures, lifestyle modifications, and specific medical interventions; a summary can be found in Table 2.

-Medical interventions:

Exogenous NO has the potential to mimic the effects of eNOS-derived NO without contributing to neurotoxicity. NO donors could be neuroprotective by increasing NO levels, enhancing capillary blood flow through vasodilation, and improving oxygen delivery to ischemic tissue [92].

Glyceryl trinitrate (GTN) functions as a nitric oxide donor and acts as both a systemic and cerebral vasodilator. The effects of transdermal GTN administration have been investigated across six randomized clinical trials involving patients who have had a stroke [93,94]. A systematic review and meta-analysis of these trials demonstrated that transdermal GTN lowers blood pressure in acute stroke settings but does not significantly influence clinical outcomes, even when administered in the ultra-early phase of stroke (≤6 h).

An individual patient data metanalysis including patients from 3 of those 7 trials evaluated the use of nitroglycerin patches in the setting of reperfusion treatment for AIS. The application of transdermal nitrates did not improve functional outcomes at 90 days, and the rates of hemorrhagic transformation were similar to those in the control group [93,94,95].

The two clinical trials investigating the prehospital administration of transdermal nitrates in patients with presumed stroke reported no benefit in reducing dependency in patients who have had a stroke and, in fact, found worse outcomes in patients with hemorrhagic stroke. These trials were RIGHT-2 (Rapid Intervention with Glyceryl Trinitrate in Hypertensive Stroke Trial-2) [96] and MR ASAP (Prehospital transdermal glyceryl trinitrate in patients with presumed acute stroke) [97].

Alternative routes for GTN administration, including intravenous infusion in the RIGID trial and intra-arterial delivery in the AGAIN trial, have also been evaluated. The RIGID trial demonstrated improvements in NIHSS score recovery, while the AGAIN trial showed reductions in infarct volume. However, the clinical significance of these findings remains limited due to the small sample sizes in both studies [98,99].

The administration of **L-arginine**, a substrate for NO synthesis, in the rat middle cerebral artery occlusion (MCAO) model has been shown to increase blood flow and reduce tissue damage [100,101]. However, some studies have reported contradictory results with L-arginine administration [102]. This discrepancy may be due to the fact that L-arginine enhances NO synthesis across all three isoforms of NOS, potentially leading to the production of neurotoxic NO by inducible iNOS or nNOS, which could counteract the neuroprotective effects of endothelial eNOS activation [103].

**BH4** is a key regulator of eNOS, and increasing vascular BH4 levels through pharmacological supplementation has been shown in experimental studies to enhance NO bioavailability [104]. In stroke-prone spontaneously hypertensive rats, intravenous infusion of BH4 significantly improved vasodilatory responses to acetylcholine [105].

Other pharmacological agents aimed at increasing NO bioavailability with preclinical evidence include DETA/NONOate [106], sodium nitroprusside [107,108], 3-morpholinosydnonimine [109,110], ZJM-289 [111], or LA-419 [112,113]. These agents have been shown to increase cell proliferation and/or reverse ischemia-induced tissue damage in the rat brain by stimulating eNOS activity while simultaneously suppressing nNOS and iNOS functions. However, clinical studies on these agents remain sparse and inconclusive.

**Edaravone**, a neuroprotective agent that acts as a free radical scavenger, including scavenging harmful NO and its oxidative metabolites, has been used to treat AIS. Administration of edaravone within 72 h of AIS has shown clinical benefits in three clinical trials. However, a Cochrane review highlighted a moderate risk of bias and small sample sizes in these studies, calling for larger, high-quality trials to confirm these findings [114].

Pharmacogenetic research found **statins** can increase eNOS expression and NO production [115]), with the effect being modulated by NOS3 polymorphisms. Statins were found to increase NOS3 mRNA levels more significantly in cultured endothelial cells with the CC genotype of the g.−786T>C polymorphism compared to those with the TT genotype [109]. Additionally, a study in hypertensive patients demonstrated that statin therapy improved cerebral endothelial function more than systemic endothelial function [116]. In this regard, there is ongoing debate about the potential increased risk of intracranial hemorrhage associated with statin use [117].

Selective inhibition of nNOS and iNOS activities is considered a promising approach for stroke treatment. N(ω)-nitro-L-arginine methyl ester (L-NAME) and 7-nitroindazole (7-NI) are NOS/nNOS inhibitors, while aminoguanidine inhibits iNOS. Preclinical evidence for these **NOS enzyme inhibitors** is well documented [118], but no clinical evidence is available from AIS patients.

-Non-pharmacological strategies

**Sphenopalatine ganglion** (SPG) **stimulation** triggers the release of neurotransmitters, including NO, which have vasodilatory effects. This approach aims to improve collateral vasodilation, stabilize the BBB, reduce edema, and promote neuroplasticity [119]. The efficacy of SPG stimulation as a therapeutic option for stroke was evaluated in a randomized, double-masked, sham-controlled trial involving 1000 patients with anterior circulation IS, but the results were inconclusive [120].

**Repetitive transcranial magnetic stimulation** (rTMS) is a non-invasive brain stimulation technique that has shown positive effects on functional recovery in patients who have had a stroke [121]. While the exact mechanisms underlying this beneficial effect are not fully understood, a study using a rat stroke model demonstrated that rTMS promoted long-term angiogenesis and reduced apoptosis in vascular EC [122].

**Ischemic preconditioning** exerts significant protective effects on endothelial function and cerebral blood flow. In focal brain ischemia models, ischemic preconditioning has been shown to enhance cerebral perfusion and penumbral blood flow, leading to improved recovery of cerebral blood flow during the post-ischemic phase [123]. These protective effects are mediated by the nitric oxide (NO) system [124]. Studies have demonstrated increased immunoreactivity of inducible nitric oxide synthase (NOS) in the cerebral vasculature 24 h after ischemic preconditioning [125]. Furthermore, the protective effects are abolished when endothelial NOS is blocked or in NOS-deficient knockout mice, underscoring its essential role [126].

Additionally, a recent study using **near-infrared laser treatment** in a mouse stroke model found that it improved cerebral blood flow and stroke outcomes, with the involvement of eNOS phosphorylation [127].

-Cell based therapies

EPC-based therapy has been explored in both animal and clinical studies. In vitro and animal models of middle cerebral artery occlusion have shown that direct **intravenous administration of EPCs** reduces stroke size and promotes beneficial compensatory mechanisms, such as restoring BBB integrity [128,129]. Recent refinements to this therapeutic approach have been tested, including virally transfecting EPCs with cytokines like adiponectin [130] or CXCL12 [131] to enhance their beneficial properties, resulting in better outcomes compared to intravenous administration alone. Additionally, the magnetic vectorization of EPCs has been evaluated, yielding promising results [132].

In humans, evidence for EPC-based therapy is still limited. A small clinical trial involving 18 patients reported slight neurological improvements in the treatment group [133]. Ongoing studies with larger sample sizes are anticipated to provide more conclusive results. However, there are significant challenges with direct intravenous infusion of EPCs, including the large number of cells required and the potential risk of tumorigenesis. To address these concerns, alternative methods of promoting EPC activity are being investigated as potentially safer and more reliable approaches. One promising approach is the use of **EPC-derived exosomes and secretomes**, which have shown encouraging results in preclinical studies, though clinical evidence in humans is still lacking [134]. Additionally, some drugs are known to promote EPC mobilization; for instance, G-CSF mobilizes EPCs from the bone marrow to peripheral blood vessels. However, a phase IIb trial did not demonstrate significant neurological improvement in patients who have had an ischemic stroke treated with G-CSF [135].

Overall, existing evidence from both preclinical and clinical studies supports the potential of EPCs as a therapeutic target, though progress in clinical applications has been slow.

-Lifestyle modification

Given the strong link between unhealthy lifestyle patterns and the deterioration of endothelial function [136], it is unsurprising that numerous epidemiological studies using a variety of methodological approaches suggest that lifestyle modifications can significantly improve endothelial function. These changes could be crucial not only in preventing vascular diseases, including stroke, but also in improving clinical outcomes.

Lifestyle changes known to improve endothelial function include regular aerobic exercise, body weight reduction, decreased salt intake, and smoking cessation.

**Physical exercise**, particularly at moderate and vigorous intensities, enhances endothelial function as measured by FMD [137,138]. During exercise, the production, bioavailability, and synthesis of NO in endothelial cells increase, resulting in the relaxation of vascular smooth muscle [138]. Recently, high-intensity interval training (HIIT) has gained popularity as an alternative to moderate-intensity exercise. HIIT is suggested to be more effective than moderate-intensity exercise in improving vascular function [139,140].

**Dietary patterns** prioritizing fruit and vegetables, whole grains, low-fat dairy, and moderate consumption of lean meats (e.g., the Mediterranean and DASH diets) can reduce the risk of vascular diseases by improving endothelial function [141,142]. A modest reduction in dietary salt intake of 3 g/day has been shown to enhance endothelial function in normotensive overweight and obese individuals. This effect may be mediated by changes in serum endothelin-1 levels [143]. A randomized cross-over study found that salt reduction improves endothelium-dependent vasodilation in normotensive subjects independently of the changes in blood pressure [144]. Dietary potassium supplement improves endothelial function with a significant increase in FMD, as shown by a meta-analysis of intervention studies [145]. Regarding vegetarian diets, the majority of the available literature suggests inefficacy in ameliorating vascular health and endothelial function markers [146]. The effect of magnesium supplementation on endothelial function was assessed in a randomized cross-over study with negative results [147].

A more comprehensive review on different types of diet and their impact on vascular health and endothelial function can be found elsewhere [148].

**Weight loss** improves endothelial-dependent vasodilation through an increased release of nitric oxide in obese hypertensive patients [149]. In this line, surgically induced weight loss with bariatric surgery significantly improved FMD that increased with time, and the resultant improvement in endothelial function was independent of weight loss or a reduction in blood pressure [150].

**Smoking cessation** leads to prolonged improvements in endothelial function, which may mediate part of the reduced cardiovascular disease risk observed after smoking cessation [151,152].

These interventions not only enhance endothelial health but also contribute to overall cardiovascular well-being, making them essential components of stroke prevention and treatment strategies.

-Combination therapies

There is growing optimism that combination therapies may offer greater efficacy in stroke treatment. Utilizing agents or approaches with different mechanisms of action simultaneously could help mitigate adverse side effects while enhancing beneficial impacts through synergistic effects. For example, combining anti-inflammatory agents with therapies targeting the NO pathway might amplify the therapeutic benefits and provide a more comprehensive approach to stroke treatment.

Previous neuroprotection trials may have failed in acute IS due to the absence of successful recanalization in treated patients. Since ischemic tissue inevitably progresses to infarction if blood flow is not restored, achieving adequate reperfusion is likely a prerequisite for recovery, with or without additional neuroprotective interventions. The four main therapeutic targets for neuroprotection are the reduction of excitotoxicity, oxidative stress, inflammation, and cellular apoptosis. In patients who achieve sufficient recanalization, an additional focus is on mitigating reperfusion injury.

With the advent of intravenous thrombolysis (IVT) and endovascular therapy (EVT), there is now an opportunity to investigate drugs with neuroprotective properties in conjunction with reperfusion therapies. However, studies that specifically assess the effects of neuroprotective agents as an adjunct to IVT and/or EVT remain limited. Future neuroprotection research should include standardized functional outcome measures and consider combining neuroprotective agents with reperfusion therapies in AIS or plan prespecified subgroup analyses for patients undergoing treatment with IVT and/or EVT.

## 6. Future Perspectives and Unanswered Questions in the Endothelial Dysfunction–Stroke Relationship

Despite recent advances in treatment options and intervention programs, stroke remains a highly prevalent and devastating condition with significant socioeconomic impact. The need for biomarkers that can support individualized stroke patient care in the context of precision medicine is becoming increasingly apparent, and the NO/endothelium pathway may offer a unique opportunity in this regard.

Ongoing research continues to explore the intricate relationship between ED and stroke, encompassing its pathophysiology, subtypes, prognosis, and response to therapies. Although considerable progress has been made, many questions remain unanswered, and several areas of uncertainty persist.

A key issue yet to be clarified is whether ED acts as an independent risk factor for stroke or if it merely reflects the influence of traditional vascular risk factors in a given patient. Consequently, it is still uncertain whether ED could be effectively leveraged to enhance personalized stroke care.

Recanalization therapies have revolutionized stroke management and prognosis, providing a new and promising scenario for testing potential therapeutic strategies. This has spurred the search for neuroprotective agents designed to maximize the benefits of ischemia-reperfusion while mitigating its associated risks.

The NO/NOS pathway plays a central role in stroke pathophysiology, offering vasodilatory and antithrombotic benefits but also posing risks such as blood–brain barrier (BBB) disruption and neurotoxicity. In the context of recanalization therapies, the NO/NOS pathway may be critical in both ischemic and reperfusion injuries, exerting effects that can be either beneficial or detrimental depending on the circumstances. Collateral response and hemorrhagic transformation are key determinants of interindividual differences in prognosis following recanalization therapies, with the NO/NOS pathway significantly influencing both, as highlighted in this review.

The impact of aging on stroke prognosis is well established, and the concept of endothelial senescence may help explain the mechanisms underlying the increased vulnerability to stroke and its treatments in older populations, opening new and promising therapeutic avenues.

Many therapeutic strategies under investigation for stroke focus on or involve the NO-endothelium pathway, with varying results. However, none have been integrated into routine clinical practice. A deeper understanding of the NO-endothelium pathway’s dual roles in different stages of the disease and, in the context of recanalization therapies, could lead to more effective clinical trials and better-guided therapeutic strategies—an important goal of this review.

Bridging the gap between scientific research and clinical practice by translating promising findings into effective preventive and therapeutic strategies for those at risk of stroke due to ED is crucial, with particular emphasis on elderly patients and those with vascular risk factors. Recent technological advancements, which have led to the development of new and enhanced research techniques, along with ongoing clinical trials, may help address some of the existing uncertainties in this field, especially concerning personalized medicine.

## Figures and Tables

**Figure 2 ijms-25-11631-f002:**
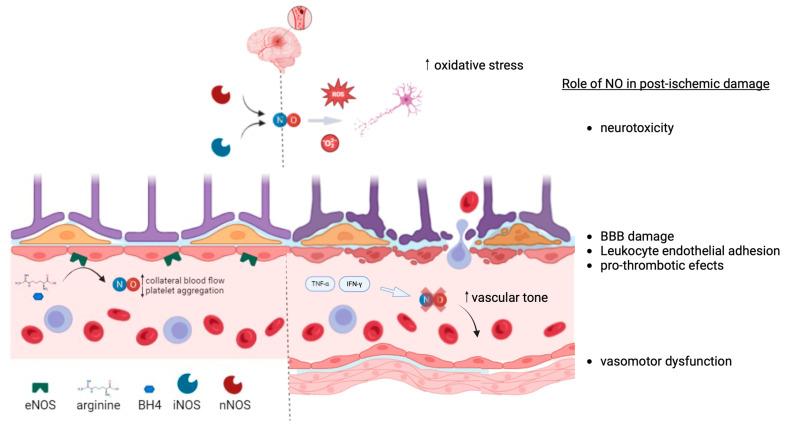
Schematic illustration of the endothelium and the NO/NOS pathway under physiological and post-ischemic conditions. In a healthy state, endothelial cells produce nitric oxide (NO) via the enzyme nitric oxide synthase (NOS), which helps maintain vascular tone, inhibit platelet aggregation, and prevent inflammation. In contrast, after ischemia, the NO/NOS pathway may become impaired, leading to reduced NO production, endothelial dysfunction, and a shift towards a proinflammatory and prothrombotic state. This comparison highlights the critical role of the NO/NOS pathway in vascular health and the potential consequences of its disruption following ischemic events. Image created in BioRender.com.

**Figure 3 ijms-25-11631-f003:**
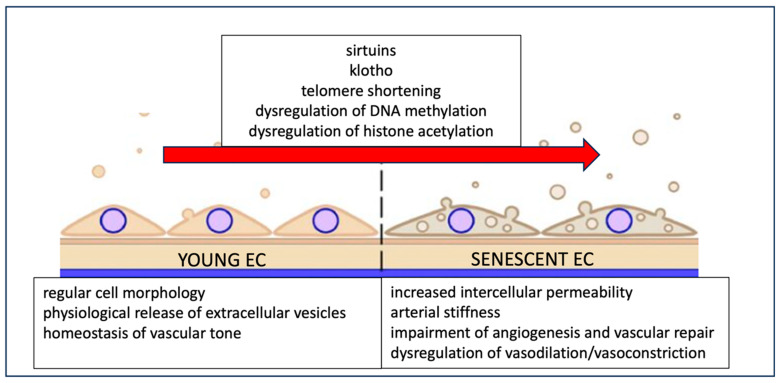
Mechanisms by which endothelial cells become senescent and their characteristics.

**Table 1 ijms-25-11631-t001:** Most commonly used techniques to assess systemic and cerebral endothelial function and studies in patients who have had a stroke.

Vascular Bed	Technique	Stimulus (Examples)	Advantages	Disadvantages	Studies in Stroke
Systemic— brachial artery	FMD	Reactive hyperemia Nytroglicerin ACh	Non-invasive inexpensive	Interobserver and intraobserver variability Lack of standarization	[37,54,56,58,59,61,62,66,67,68]
Systemic— microvascular circulation	Disposable pletismographic probes	Reactive hiperemia Capsaicin	Reproducibility Easy, automated	Expensive	[66]
	Venous occlusion pletismography	Ach	Contralateral arm as control Dose response relationship	Invasive Time consuming	--
Cerebral circulation	Transcranial doppler sonography	L-Arginine l-NMA Acetazolamide	Inexpensive Reproducible Availability	CO_2_ dependent Intravenous infusion Adverse events to drugs	[49,58,59,60,62]
	PET, SPECT, Xe-CT, perfusion CT, perfusion MRI	L-Arginine l-NMA Acetazolamide	Quantitative Regionally specific information	Expensive Not widely accesible	--
Coronary circulation	Doppler wires	Ach Adenosine Papaverine	Direct assessment of the coronary circulation	Invasive Expensive Time intensive Limited to those undergoing coronary angiography	--

Ach: acetylcholine; l-NMA: NG-methyl-L-arginine; FMD: flow mediated dilation; CT: cranial tomography; MRI: magnetic resonance imaging.

**Table 2 ijms-25-11631-t002:** Summary of the therapeutic approaches in the field of the NO endothelium pathway and stroke.

Medical Interventions	Non-Pharmacological Strategies	Cell Based Therapies	Lifestyle Modifications
Glyceryl trinitrate	Sphenopalatine ganglion (SPG) stimulation	Administration of EPCs	Physical exercise
L-arginine	Repetitive transcranial magnetic stimulation	EPC-derived exosomes and secretomes	Dietary patterns
BH4	Ischemic preconditioning		Salt reduction
Statins	Near-infrared laser treatment		Body weight reduction
NOS enzyme inhibitors			Smoking cessation

## Data Availability

Not applicable.

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
