# Peer review of "Understanding Endothelial Dysfunction and Its Role in Ischemic Stroke After the Outbreak of Recanalization Therapies"

_ijms, 2024, doi:10.3390/ijms252111631_

Round 1

Reviewer 1 Report

Comments and Suggestions for Authors

In this review, the authors underlying the advances in stroke treatment options, stroke remains a highly prevalent and devastating condition with significant socioeconomic impact. The authors aim to synthesize current evidence on the role of the nitric oxide (NO)/endothelium pathway in ischemic stroke. Despite recent advances in understanding the link between endothelial dysfunction and stroke, many uncertainties remain. Although treatments targeting this pathway show promise, they have yet to yield definitive clinical benefits.

Questions:

1. Introduction: 

insert some epidemiological data

2. The endothelium and its components: 

-       insert a scheme, to clarify the role of NO, eNOS, VEGF….in endothelium and its components 

2.2.  Nitric oxide ïƒ  expand the paragraph

5. Therapeutic approaches regarding endothelial dysfunction and stroke: 

the whole paragraph needs to be revised. Insert recently bibliography and explain better the non pharmacological strategies. It might be useful to insert some tables, diagrams, schemes, etc

Comments on the Quality of English Language

none

Author Response

Reviewer 1  

  1. Introduction: 
  • insert some epidemiological data:
    • Agree, we have added epidemiological data as suggested:” According to the most recent Global Burden of Disease (GBD) estimates, in 2021, stroke was the third most common GBD level 3 cause of death (7·3 million [95% UI 6·6–7·8] deaths; 10·7% [9·8–11·3] of all deaths), and the fourth most common cause of DALYs (160·5 million [147·8–171·6] DALYs; 5·6% [5·0–6·1] of all DALYs). In 2021, there were 93·8 million (89·0–99·3) prevalent and 11·9 million (10·7–13·2) incident strokes”.

Page 2, lines 61-65.

  1. The endothelium and its components: 

-       Insert a scheme, to clarify the role of NO, eNOS, VEGF….in endothelium and its components.

  • A figure has been added with this purpose, figure 2. Subsequent figures have been renamed. Page 6, line 265.

2.2.  Nitric oxide à expand the paragraph

  • Agree, we have expanded the paragraph about Nitric oxide as follows:

“Nitric oxide (NO) is often considered a toxic gas in nature. However, 40 years ago it was discovered to function as a signaling molecule, particularly in the endothelium, where it acts as a primary neurotransmitter. NO is synthesized from L-arginine through the mediation of the NOS enzyme, with tetrahydrobiopterin (BH4) required as a cofactor. Once produced, NO activates guanylyl cyclase, which leads to the production of cGMP and the subsequent activation of various proteins essential for brain function. The fast metabolism and brief half-life of NO make it challenging to measure in clinical environments. Nitrite and nitrate, the byproducts of NO autoxidation, have been utilized as indicators of NO production.[9].

Nitric oxide (NO) regulates vascular smooth muscle tone and inhibits platelet aggregation, leukocyte adhesion, and vascular smooth muscle cell growth [10]. It is widely acknowledged that the loss of endothelial NO is a central mechanism in the pathogenesis of endothelial dysfunction (ED). In both cerebral and peripheral vasculatures, reduced availability of endothelial NO leads to significant detrimental alterations in vascular functions[11].

Reduced endothelial NO availability has been demonstrated to encourage a proinflammatory and prothrombotic endothelial phenotype. NO bioavailability is closely linked to cardiovascular health, and it has been proposed that variations in NO bioavailability may help differentiate individuals with good vascular health—who are less likely to experience cardiovascular events—from those with poor vascular health[12]

Page 3, line 118

  1. Therapeutic approaches regarding endothelial dysfunction and stroke: 

the whole paragraph needs to be revised. Insert recently bibliography and explain better the non pharmacological strategies. It might be useful to insert some tables, diagrams, schemes, etc

  • Thank you for pointing this out, the whole paragraph has been revised, we have expanded the paragraphs about medical interventions and particularly the non- pharmacological approaches sections. In addition, more recent references have been added a summarizing table, table 2, has been included in the end of the section.

Page 13, lines 560-784

Reviewer 2 Report

Comments and Suggestions for Authors

1. The introduction provides a good overview of the topic, but consider briefly introducing the relationship between endothelial dysfunction and recanalization therapies early in the introduction to set the context.

2. Section 4 (Endothelial Dysfunction and Stroke): This section could benefit from clearer subheadings that distinguish between the role of endothelial dysfunction in stroke initiation versus progression and prognosis. The connection between reperfusion therapies and endothelial dysfunction could be highlighted more prominently.

3. In Table 1, where you present techniques for assessing endothelial function, ensure that all abbreviations and terms are defined clearly for readers unfamiliar with them.

4. The discussion on unanswered questions (section 6) is strong. However, you could further expand on how newer techniques or ongoing clinical trials might resolve some of the uncertainties, particularly in the area of personalized medicine.

5. The reference style appears to be consistent, but ensure that all citations are properly formatted according to the journal's guidelines.

Comments on the Quality of English Language

Minor editing of English language required.

Author Response

  1. The introduction provides a good overview of the topic, but consider briefly introducing the relationship between endothelial dysfunction and recanalization therapies early in the introduction to set the context.

- Agree. We have therefore added this paragraph about it in the introduction:

“Recanalization therapies, including the intravenous administration of fibrinolytic agents and endovascular thrombectomy, are the most effective proven treatments for IS. Endothelial dysfunction can negatively affect the success of recanalization therapies, which are used to restore blood flow in stroke, by increasing reperfusion injury, compromising the blood-brain barrier, impairing vasodilation, promoting a prothrombotic state, and enhancing inflammatory responses”.

Page 2, line 76

  1. Section 4 (Endothelial Dysfunction and Stroke): This section could benefit from clearer subheadings that distinguish between the role of endothelial dysfunction in stroke initiation versus progression and prognosis. The connection between reperfusion therapies and endothelial dysfunction could be highlighted more prominently.

- Thank you for this comment, we have added the proposed subheadings in this section.

 Page 7, line 354

  • We also have remarked the connection between reperfusion therapies and endothelial dysfuction in a separate pararagraph as follows:“Endothelial dysfunction could play a role in the effectiveness and outcomes of recanalization therapies. First, when blood flow is restored to previously ischemic tissue, it can trigger reperfusion injury, which is characterized by inflammation and oxidative stress. Endothelial dysfunction can exacerbate this injury by contributing to the breakdown of the blood-brain barrier, increased production of ROS , and heightened inflammatory responses This can worsen tissue damage even after successful recanalization [80]. Second, a healthy endothelium maintains the integrity of the BBB and when ED is present, the BBB may be compromised, increasing the risk of HT and cerebral edema following recanalization[81]. Third, In ED, reduced NO bioavailability can impair vasodilation, limiting the effectiveness of recanalization therapies as adequate NO levels are necessary for optimal blood flow restoration to ischemic regions. Fourth, ED is associated with a prothrombotic state, increasing the risk of reocclusion after recanalization[5]. Fifth, thrombus removal using stent retrievers is associated with gadolinium vessel enhancement and BBB disruption in about 50% of patients, suggesting that local endothelial damage increases with the number of stent passes and the use of alteplase[82]. And last, half of the thrombi retrieved from patients treated with stent retrievers contain endothelial cells, indicating endothelial damage secondary to direct mechanical injury from thrombectomy[83]”.

Page 11 line 460

  1. In Table 1, where you present techniques for assessing endothelial function, ensure that all abbreviations and terms are defined clearly for readers unfamiliar with them.

  • We have reviewed the table and one of the abbreviations was incomplete and we have corrected it.

Page 10, line 422.

  1. The discussion on unanswered questions (section 6) is strong. However, you could further expand on how newer techniques or ongoing clinical trials might resolve some of the uncertainties, particularly in the area of personalized medicine.

- Thank you for pointing this out. We have added a sentence about it at the end of section 6: “Recent technological advancements, which have led to the development of new and enhanced research techniques, along with ongoing clinical trials, may help address some of the existing uncertainties in this field, especially concerning personalized medicine”.

Page 17 line 813

  1. The reference style appears to be consistent, but ensure that all citations are properly formatted according to the journal's guidelines.

            - We have revised the format of the references as proposed

Round 2

Reviewer 1 Report

Comments and Suggestions for Authors

The authors answered my questions

the work improved.

Thanks

Comments on the Quality of English Language

none